# GPS-Net: Graph-based Photometric Stereo Network

**Zhuokun Yao[1], Kun Li[1*], Ying Fu[2], Haofeng Hu[3], Boxin Shi[4,5*]**
[1]College of Intelligence and Computing, Tianjin University, Tianjin, China
[2]School of Computer Science and Technology,
Beijing Institute of Technology, Beijing, China
[3]School of Precision Instrument and Opto-Electronics Engineering,
Tianjin University, Tianjin, China
[4]Department of Computer Science and Technology, Peking University, Beijing, China
[5]Institute for Artificial Intelligence, Peking University, Beijing, China
{yaozk,lik,haofeng_hu}@tju.edu.cn, fuying@bit.edu.cn, shiboxin@pku.edu.cn

## Abstract

Learning-based photometric stereo methods predict the surface normal either in a per-pixel or an all-pixel manner. Per-pixel methods explore the inter-image intensity variation of each pixel but ignore features from the intra-image spatial domain. All-pixel methods explore the intra-image intensity variation of each input image but pay less attention to the inter-image lighting variation. In this paper, we present a Graph-based Photometric Stereo Network, which unifies per-pixel and all-pixel processings to explore both inter-image and intra-image information. For per-pixel operation, we propose the Unstructured Feature Extraction Layer to connect an arbitrary number of input image-light pairs into graph structures, and introduce Structure-aware Graph Convolution filters to balance the input data by appropriately weighting shadows and specular highlights. For all-pixel operation, we propose the Normal Regression Network to make efficient use of the intra-image spatial information for predicting a surface normal map with rich details. Experimental results on the real-world benchmark show that our method achieves excellent performance under both sparse and dense lighting distributions.

## 1 Introduction

Photometric stereo aims at estimating surface normals of a static object from a set of images acquired under various illumination conditions from a fixed camera [1]. The pixel-wise estimation of surface orientation makes photometric stereo outstanding in acquiring high-resolution 3D information. Recent progress in deep learning has been verified to be effective in photometric stereo for general complex reflectance, showing superior accuracy over conventional methods on a benchmark dataset [2]. To design effective deep learning frameworks in the context of photometric stereo, the core problem is how to deal with a sequence of unordered and arbitrary numbers of input images under various illumination conditions. The first deep photometric stereo network [3] fixes the order and number of input images during training and testing. Following works focus on relaxing this unpractical assumption in dealing with unordered, arbitrary numbers of images, either in a per-pixel [4]–[6] or an all-pixel [7]–[9] manner, *i.e.*, whether the observations of a single pixel or the whole image are fed to the network, according to a recent survey about data-driven photometric stereo [10].

As illustrated in the top row of Fig. 1, per-pixel methods explore the *inter-image* intensity[1] variation by projecting the observations of each pixel into a fixed-size observation map, according to the first

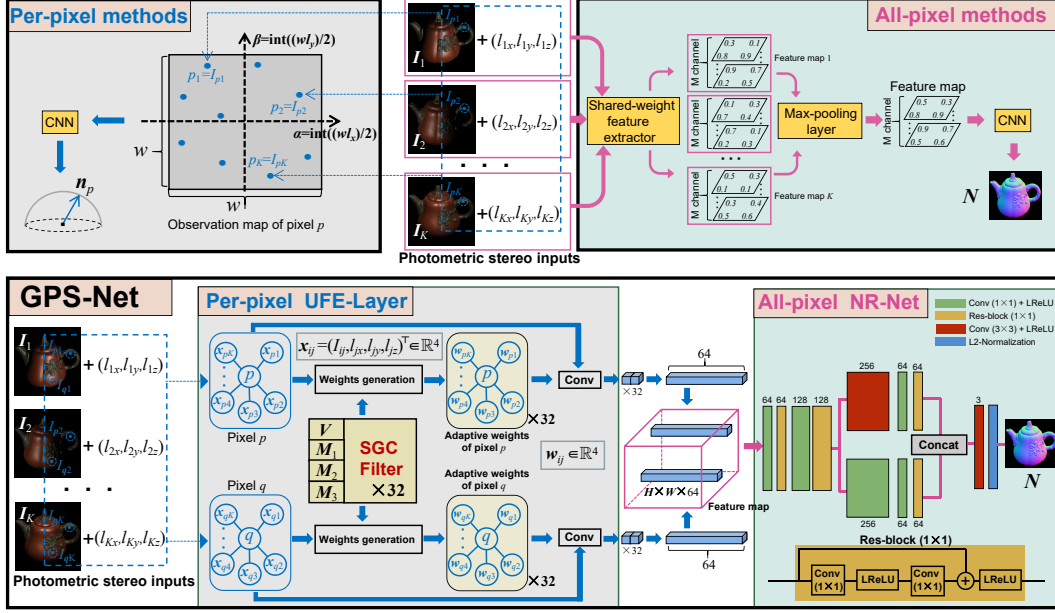

Figure 1: Illustration of GPS-Net (bottom row) and state-of-the-art per-pixel [4]–[6] and all-pixel [7], [8] learning-based photometric stereo methods (top row).

two components of the two-degree-of-freedom normalized lighting directions ($l_x$ and $l_y$) [4]–[6]. All-pixel methods explore the *intra-image* intensity variation by extracting features of each image-light pair separately using a shared-weight feature extractor before fusing them using a size-independent max-pooling layer [7], [8]. Then the unordered, arbitrary numbers of inputs are converted into a structured observation map or feature map, which can be fed into the subsequent CNN-based network to regress a pixel-wise surface normal or a complete normal map. Per-pixel methods ignore features from the intra-image spatial domain, and their observation maps have to take a trade off between resolution and density, which makes them difficult to maintain good performance when the number of input images varies from sparse to dense. In contrast, the independent processing for each image in all-pixel methods prevents them from exploring the inter-image lighting variation, and their networks that make extensive use of 3×3 convolutional layers cause over-smoothing and loss of resolution details in the spatial domain.

In this paper, we present an end-to-end Graph-based Photometric Stereo Network, namely GPS-Net, which combines the advantages of per-pixel and all-pixel methods to explore both inter-image and intra-image variation, as shown in the bottom row of Fig. 1. To explore per-pixel information, the Unstructured Feature Extraction Layer (UFE-Layer) is designed to connect an arbitrary number of inter-image observations of each pixel into a graph structure to avoid introducing the problem of the density of valid data; the Structure-aware Graph Convolution (SGC) filters in UFE-Layer are then adopted to deal with the topologically inconsistent graphs and extract a fixed-size feature map from the unstructured data. SGC filters also learn adaptive weights for suppressing outliers (*e.g.*, attached shadows and cast shadows), and emphasizing useful observations (*e.g.*, specular highlights). To explore all-pixel information, the Normal Regression Network (NR-Net) is further designed to make efficient use of the intra-image spatial information and regress a high-resolution and high-accuracy normal map. Experimental results demonstrate that GPS-Net achieves superior performance over state-of-the-art per-pixel and all-pixel methods, provides stable performance from sparse to dense lighting distributions, and maintains rich surface normal details for each pixel.

## 2 Related Work

### 2.1 Conventional Methods

Under the ideal Lambertian reflectance assumption, the classic photometric stereo algorithm [1] estimates surface normals through a least-squares method. To extend the classic Lambertian algorithm to more practical real-world scenes, many Non-Lambertian methods are proposed.

Outlier rejection methods treat Non-Lambertian observations as sparse outliers, and reject them through RANSAC [11], taking median values [12], expectation maximization [13], or sparse Bayesian regression [14]. Model-based methods such as Torrance-Sparrow model [15], Ward model [16], Cook-Torrance model [17], specular spike model [18], [19], microfacet model [20], and empirical model [21]–[27] are proposed to approximate analytic Non-Lambertian Bidirectional Reflectance Distribution Functions (BRDFs). Example-based methods [28], [29] guide normal estimation by taking advantage of the observations of a known reference object captured under the same lighting condition as the target object. Please refer to the survey in [2] for more detailed discussions.

## 2.2 Learning-based Methods

Recently, with the progress of deep learning, several learning-based approaches [3]–[8], [30], [31] are proposed and achieve excellent performance. The major difficulty of learning-based photometric stereo frameworks lies in how to deal with an arbitrary number of unstructured inputs. DPSN [3] and OUTDOOR-PS [31] take a fixed light source setting or fixed number of lights to simplify the unstructured inputs, which makes them impractical for randomly distributed lightings with sparse to dense variation. To relax this inflexible assumption and deal with diverse distributions of lightings, state-of-the-art learning-based frameworks adopt two strategies — per-pixel methods [4]–[6] using the observation map strategy, and all-pixel methods [7], [8] using the mechanism of shared-weight feature extractor and max-pooling layer, based on a recent survey in [10].

CNN-PS [4] generates a fixed-size observation map that merges all the observations of each pixel, from which a CNN-based network is further used to predict a surface normal in a per-pixel manner. Due to the contradiction between the density and resolution of the observation map, CNN-PS has difficulty in achieving good performance under sparse and dense lighting distributions at the same time, and its pixel-wise processing cannot explore the intra-image spatial information. LMPS [5] and SPLINE-Net [6] are proposed to solve sparse photometric stereo following the observation map strategy. PS-FCN [7] and SDPS-Net [8] aggregate features extracted by a shared-weight feature extractor using a size-agnostic max-pooling layer and then feed them to a CNN-based network to estimate a complete normal map. However, they pay less attention to the inter-image lighting variation. In addition to frameworks trained in a supervised manner, Taniai and Maehara [30] propose an unsupervised all-pixel photometric stereo network. However, it has the drawback of time-consuming computation. Please refer to the survey in [10] for more details.

## 3 Problem Formulation

Calibrated photometric stereo takes a set of intensity observations $I$ under directionally varying lightings $\boldsymbol{l} = (l_x, l_y, l_z)^\top \in \mathbb{R}^3$ as input, and outputs the estimated surface normal $\boldsymbol{n}$.[2] For per-pixel photometric stereo methods, $I$ and $\boldsymbol{n}$ are defined for each pixel, which can be represented as $I = I_i \in \mathbb{R}$ and $\boldsymbol{n} = \boldsymbol{n}_i = (n_{ix}, n_{iy}, n_{iz})^\top \in \mathbb{R}^3$, where $i$ indicates the pixel index. For all-pixel methods, $I$ and $\boldsymbol{n}$ are usually represented in matrix format as $I = \boldsymbol{I} \in \mathbb{R}^{H \times W}$ and $\boldsymbol{n} = \boldsymbol{N} \in \mathbb{R}^{H \times W \times 3}$, where $H$ and $W$ indicate the height and width of the input image. Given a sequence of inputs $\{[I, \boldsymbol{l}]\} = \{[I_j, \boldsymbol{l}_j] | 1 \leq j \leq K\}$, where $j$ indicates that the observation is captured under the illumination of the $j$-th light source and $K$ is the number of lights, $\boldsymbol{n}$ can be calculated by inversely solving the image formation model via

$$I = \rho(\boldsymbol{n}, \boldsymbol{l}, \boldsymbol{v}), \tag{1}$$

where $\boldsymbol{v} = (0, 0, 1)^\top$ represents the outgoing viewing direction, and the BRDF model is usually used to describe the function $\rho$. Unlike conventional methods inversely solving Eq. (1), learning-based approaches directly learn the mapping from $\{[I, \boldsymbol{l}]\}$ to $\boldsymbol{n}$.

The state-of-the-art learning-based methods can be divided into per-pixel methods [4]–[6] and all-pixel methods [7], [8]. For per-pixel methods, their input can be represented as $\{[I, \boldsymbol{l}]\} = \{[I_{ij}, \boldsymbol{l}_j] | 1 \leq j \leq K\}$. As illustrated in Fig. 1, these methods predict a surface normal for pixel $i$ via

$$\boldsymbol{n} = \boldsymbol{n}_i = \text{CNN}_{\text{per}}(\text{OBS}(\{[I_{ij}, \boldsymbol{l}_j] | 1 \leq j \leq K\})), \tag{2}$$

where OBS indicates that all the inter-image information is projected into an observation map.

For all-pixel methods, their input can be represented as $\{[I, l]\} = \{[\boldsymbol{I}_j, \boldsymbol{l}_j] | 1 \leq j \leq K\}$. They predict a complete normal map via

$$\boldsymbol{n} = \boldsymbol{N} = \text{CNN}_{\text{all}}(\max(\{\text{SFE}([\boldsymbol{I}_j, \boldsymbol{l}_j]) | 1 \leq j \leq K\})), \tag{3}$$

where SFE stands for the shared-weight feature extractor. It can be seen from Eq. (2) and Eq. (3) that per-pixel methods do not explicitly extract features from the intra-image spatial domain, and all-pixel methods pay less attention to the inter-image information that reflects the pixel-wise variation under different lightings. Unlike per-pixel and all-pixel approaches that focus only on one "dimension" of photometric stereo input, we design a network that extracts both inter-image and intra-image features. As illustrated in Fig. 1, our method can be formulated as

$$\boldsymbol{n} = \boldsymbol{N} = \text{CNN}(\{\text{SGC}(\{[I_{ij}, \boldsymbol{l}_j] | 1 \leq j \leq K\}) | 1 \leq i \leq H \times W\}), \tag{4}$$

where SGC represents the proposed SGC filter.

## 4 Graph-based Photometric Stereo Network

In this section, we introduce our Graph-based Photometric Stereo Network, called GPS-Net. As shown in Fig. 1, GPS-Net consists of two modules, *i.e.*, the UFE-Layer for per-pixel operation to connect the unstructured inter-image inputs of each pixel into a graph structure and generate a fixed-size feature map, and the NR-Net for all-pixel operation to further regress a normal map from the intra-image spatial information.

### 4.1 UFE-Layer for Per-pixel Operation

To deal with the unstructured inter-image information of each pixel, previous per-pixel learning-based methods [4]–[6] adopt an observation map to aggregate an arbitrary number of pixel-wise observations. However, this strategy has two problems. First, as shown in Fig. 1, the size $w$ of the observation map is fixed. If $w$ is too small, the resolution of the observation map will be reduced so that its performance will drop significantly. If $w$ is too large, the valid data can only occupy a small proportion of the observation map, causing the sparsity problem and poor performance. This contradiction between resolution and density makes per-pixel approaches difficult to find a balance between sparse and dense lighting distributions. Second, observation maps cannot retain the intra-image spatial information that can be further used to improve performance. To solve these two problems, we propose the UFE-Layer. Moreover, an inherent advantage of per-pixel methods is that observation maps reflect the distribution of outliers in their spatial domain [4]–[6]. Our UFE-Layer can also distinguish negative and positive observations through adaptive weighting for attached/cast shadow outliers and specular highlight clues using the proposed SGC filters.

As shown in Fig. 1, UFE-Layer connects an arbitrary number of inter-image observations of each pixel ($\{[I_{ij}, \boldsymbol{l}_j] | 1 \leq j \leq K\}$) into a graph structure. The central node that indicates the pixel index does not carry any information, and each adjacent node contains the input information under a certain lighting ($\boldsymbol{x}_{ij} = [I_{ij}, \boldsymbol{l}_j] = (I_{ij}, l_{jx}, l_{jy}, l_{jz})^\top \in \mathbb{R}^4$). The number of adjacent nodes equals to the number of lights. Since the number of lights may vary from sparse to dense, the graph structure also has an arbitrary number of adjacent nodes. To efficiently extract features from an unstructured graph with arbitrary numbers of adjacent nodes, we design the SGC filters, which are inspired by the convolution strategy for topologically inconsistent graphs [32]. Each SGC filter is composed of four learnable parameter matrices, *i.e.*, $\boldsymbol{V} = (\boldsymbol{v}_1, \boldsymbol{v}_2, \boldsymbol{v}_3, \boldsymbol{v}_4) = ((v_{11}, v_{12}, ..., v_{1t})^\top, ..., (v_{41}, v_{42}, ..., v_{4t})^\top) \in \mathbb{R}^{t \times 4}$, $\boldsymbol{M}_1 \in \mathbb{R}^{4 \times s}$, $\boldsymbol{M}_2 \in \mathbb{R}^{s \times s}$, and $\boldsymbol{M}_3 \in \mathbb{R}^{s \times 1}$, where $t$ and $s$ are hyperparameters, and 4 is the number of channels of $\boldsymbol{x}_{ij}$. The SGC filter generates diverse weights $\boldsymbol{w}_{ij} = (w_{ij1}, w_{ij2}, w_{ij3}, w_{ij4})^\top$ for each adjacent node so that we can convolve $\boldsymbol{x}_{ij}$ through an inner product operation to get a 2-channel output feature vector of pixel $i$, which can be formulated as

$$\boldsymbol{y}_i = (\ \frac{1}{K} \sum_{j=1}^{K} (\boldsymbol{x}_{ij} \cdot \boldsymbol{w}_{ij}),\ \max(\{\boldsymbol{x}_{ij} \cdot \boldsymbol{w}_{ij} | 1 \leq j \leq K\})\ )^\top \in \mathbb{R}^2. \tag{5}$$

There are two points worth noting. First, we adopt an averaging operation instead of a summing operation that is widely used in graph convolution strategies, which aims at making our network more robust to the drastically varying number of photometric stereo inputs ($K$). Second, we further expand

the feature vector by introducing the maximum convolution result, which has been proven to provide strong clues for surface normal inference [7].

To generate diverse weights, we first use three learnable parameter matrices and three nonlinear operations to quantitatively calculate the topological relationship $r_{ij}$ of each adjacent node according to the information it carries via

$$r_{ij} = \mathrm{Tanh}(\mathrm{ReLU}(\mathrm{ReLU}(\boldsymbol{x}_{ij}^{\top}\boldsymbol{M}_1)\boldsymbol{M}_2)\boldsymbol{M}_3). \tag{6}$$

Then, we take $r_{ij}$ as the argument of an adaptively fitted weighting function and use the function value to represent the generated weight, which can be formulated as

$$w_{ijc} = \sum_{k=1}^{t} v_{ck} \cdot h_k(r_{ij}), \tag{7}$$

where the learnable coefficient $v_{ck}$ is from matrix $\boldsymbol{V}$. $\boldsymbol{v}_c = (v_{c1}, v_{c2}, ..., v_{ct})^{\top}$ $(1 \leq c \leq 4)$ is used to fit a weighting function for the $c$-th channel with Chebyshev polynomials truncated to $t$ terms. $h_k(\cdot)$ expresses the $k$-term Chebyshev polynomial which can be generated by the stable recurrence relation, *i.e.*, $h_k(r) = 2rh_{k-1}(r) - h_{k-2}(r)$, with $h_1(r) = 1$ and $h_2(r) = r$. Hyperparameter $t$ affects the expressive ability of the fitted weighting functions, and hyperparameter $s$ affects the performance of the quantitative calculation of topological relationship. We experimentally find that $t = 32$ and $s = 10$ can guarantee a sufficiently good performance, while a larger setting will increase the number of network parameters without bringing performance improvement. Finally, we use 32 SGC filters so that each pixel can generate a 64-channel feature vector. All the feature vectors are arranged together according to the pixel position to obtain a structured 64-channel feature map.

The UFE-Layer explores the unstructured inter-image information without suffering from the contradiction between resolution and density in the observation map [4]. Therefore, we can achieve consistently good performance under both sparse and dense lightings. The UFE-Layer also retains the intra-image spatial information used by the subsequent NR-Net to improve prediction accuracy.

## 4.2 NR-Net for All-pixel Operation

After processing the input data using the UFE-layer pixel-wisely, the unstructured input is transformed into a structured $H \times W \times 64$-dimensional feature map with its spatial information preserved. We further regress a complete normal map using the NR-Net by operating in an all-pixel manner. Existing all-pixel learning-based methods [7], [8] use $3 \times 3$ filters in all convolutional layers in their networks to predict a normal map. This brings a problem. Although $3 \times 3$ convolutional layers can improve performance by exploring the intra-image spatial information, too many $3 \times 3$ convolutional layers will cause over-smoothing in the spatial domain, leading to the loss of resolution details and poor performance. In contrast, a moderate number of $3 \times 3$ convolutional layers can find a balance between over- and proper- smoothing spatial information to improve prediction accuracy while preserving high-resolution image details. To this end, we adopt a multi-scale and multi-branch design in the NR-Net, as shown in Fig. 1. Unlike all-pixel methods with all-$3 \times 3$ architectures, we adopt an appropriate number of $3 \times 3$ convolutional layers to make efficient use of the intra-image spatial information, which avoids the blurring effect and performance degradation caused by over-smoothing. To preserve the pixel-wise resolution details to the greatest extent, we further divide NR-Net into two branches with $1 \times 1$ and $3 \times 3$ architectures to ensure that there is an all-$1 \times 1$ branch operating in the pixel-wise manner before the final convolutional layer. Experimental results in Section 5.4 prove that NR-Net can restore image details better than all-pixel methods, and experimental results in Section 5.5 verify that our multi-scale and multi-branch design achieves the best prediction accuracy.

## 4.3 Loss Function

The learning of our GPS-Net is supervised by the angular error loss function via

$$L_{AE} = \frac{1}{P} \sum |\arccos(\boldsymbol{n}_p \cdot \boldsymbol{n}_p')|, \tag{8}$$

where $p$ accounts for all the pixels located in the foreground mask, and $P$ is the number of them. $\boldsymbol{n}_p$ and $\boldsymbol{n}_p'$ represent the predicted surface normal and the ground truth, respectively. Other losses like mean square error [4], [5], [30] and cosine similarity error [7], [8] can also be utilized.

Table 1: Performance on the DiLiGenT benchmark with different numbers of input images. We perform the testing on all the ten objects contained in the DiLiGenT dataset and average the results. The values represent MAEs in degree (the lower the better). The penultimate column represents the average performance of each method, and the last column represents their standard deviations. **Black bold** texts indicate the best performance, and underlined texts indicate the second best.

| | Number of input images | | | | | | | Avg. | Std. |
|---|---|---|---|---|---|---|---|---|---|
| | 4 | 8 | 10 | 16 | 32 | 64 | 96 | | |
| LS [1] | 18.79 | 16.36 | 16.10 | 15.73 | 15.51 | 15.42 | 15.39 | 16.19 | **1.12** |
| CNN-PS [4] | 47.82 | 18.44 | 13.53 | 10.40 | 8.18 | **7.56** | **7.21** | 16.16 | 13.45 |
| LMPS [5] | 15.61 | 10.39 | 10.01 | 9.66 | 9.38 | 9.15 | 8.41 | 10.37 | 2.22 |
| SPLINE-Net [6] | 17.05 | 11.32 | 10.35 | 10.12 | 9.93 | 9.72 | 9.63 | 11.16 | 2.46 |
| NEURAL-PS [30] | 16.86 | 11.57 | 10.79 | 9.87 | 9.38 | 8.98 | 8.83 | 10.90 | 2.60 |
| PS-FCN [7] | 16.50 | 10.84 | 10.19 | 9.20 | 8.74 | 8.47 | 8.39 | 10.33 | 2.66 |
| Ours | **13.46** | **10.07** | **9.43** | **8.71** | **8.05** | 7.84 | 7.81 | **9.34** | 1.86 |

Table 2: Results on each object in the DiLiGenT benchmark with 10 and 96 input images.

| | 10 input images | | | | | | | | | | |
|---|---|---|---|---|---|---|---|---|---|---|---|
| | ball | cow | bear | cat | pot1 | pot2 | buddha | goblet | reading | harvest | Avg. |
| LS [1] | 4.58 | 26.48 | 9.84 | 8.90 | 9.59 | 15.65 | 16.02 | 19.23 | 19.37 | 31.32 | 16.10 |
| CNN-PS [4] | 8.21 | 13.83 | 11.89 | 9.00 | 12.79 | 15.04 | 13.39 | 15.74 | 16.07 | 19.36 | 13.53 |
| LMPS [5] | 3.97 | 10.19 | 8.73 | 6.69 | 7.30 | 9.74 | 11.36 | 10.46 | 14.37 | 17.33 | 10.01 |
| SPLINE-Net [6] | 4.96 | **8.80** | 5.99 | 7.52 | 8.77 | 11.79 | 10.07 | **10.43** | 16.13 | 19.05 | 10.35 |
| NEURAL-PS [30] | **2.12** | 8.87 | 6.92 | **6.58** | **7.14** | 9.61 | 11.41 | 14.99 | **13.70** | 26.55 | 10.79 |
| PS-FCN [7] | 4.35 | 9.97 | **5.70** | 8.24 | 8.38 | 10.37 | 10.54 | 11.21 | 14.34 | 18.82 | 10.19 |
| Ours | 4.33 | 9.34 | 6.34 | 6.81 | 7.50 | **8.38** | 8.87 | 10.79 | 15.00 | **16.92** | **9.43** |

| | 96 input images | | | | | | | | | | |
|---|---|---|---|---|---|---|---|---|---|---|---|
| | ball | cow | bear | cat | pot1 | pot2 | buddha | goblet | reading | harvest | Avg. |
| LS [1] | 4.10 | 25.60 | 8.39 | 8.41 | 8.89 | 14.65 | 14.92 | 18.50 | 19.80 | 30.62 | 15.39 |
| CNN-PS [4] | 2.12 | 7.92 | **4.20** | **4.38** | **5.37** | **6.38** | 8.07 | **7.42** | 12.12 | **14.08** | **7.21** |
| LMPS [5] | 2.40 | 7.98 | 5.23 | 6.11 | 6.54 | 7.48 | 9.89 | 8.61 | 13.68 | 16.18 | 8.41 |
| SPLINE-Net [6] | 4.51 | 7.44 | 5.28 | 6.49 | 8.29 | 10.89 | 10.36 | 9.62 | 15.50 | 17.93 | 9.63 |
| NEURAL-PS [30] | **1.47** | 6.32 | 5.79 | 5.44 | 6.09 | 7.76 | 10.36 | 11.47 | **11.03** | 22.59 | 8.83 |
| PS-FCN [7] | 2.82 | 7.33 | 7.55 | 6.16 | 7.13 | 7.25 | 7.91 | 8.60 | 13.33 | 15.85 | 8.39 |
| Ours | 2.92 | **6.14** | 5.07 | 5.42 | 6.04 | 7.01 | **7.77** | 9.00 | 13.58 | 15.14 | 7.81 |

# 5  Experimental Results

## 5.1  Datasets and Settings

**Synthetic dataset for the training.**  For the training, we use the synthetic photometric stereo dataset made by Chen *et al.* [7], which renders shapes from the Blobby shape dataset [33] and the Sculpture shape dataset [34] with the MERL BRDF dataset [35]. This dataset contains 85212 samples rendered under 64 directional lightings, which are randomly split into 99:1 for training and validation.

**Synthetic dataset for the testing.**  To test our model on different materials, we rendered a "sphere" and a "bunny" with all the 100 BRDFs from the MERL dataset [35] under the illumination of 100 randomly distributed light sources (sampled from a range of $180° \times 180°$).

**Real datasets for the testing.**  For the quantitative comparison, we use the DiLiGenT dataset [2] which is a real-world public benchmark containing images of 10 objects with complex reflectance illuminated by 96 directional lightings. We further perform qualitative experiments on two challenging real datasets without ground truth, *i.e.*, the Light Stage Data Gallery dataset [36] containing 6 objects illuminated by 253 directional lightings, and the Gourd&Apple dataset [22] containing 3 objects illuminated by 112, 102, and 98 directional lightings, respectively.

Table 3: Results on the synthetic dataset with different materials. We perform the testing on all the 100 BRDFs and average the results.

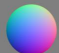

| Sphere  Bunny | Sphere | | | | Bunny | | | |
| --- | --- | --- | --- | --- | --- | --- | --- | --- |
| | Number of inputs | | | | Number of inputs | | | |
| | 4 | 32 | 100 | Avg. | 4 | 32 | 100 | Avg. |
| CNN-PS [4] | 45.01 | 5.14 | **3.14** | 17.76 | 42.38 | 6.00 | **4.11** | 17.50 |
| PS-FCN [7] | 17.96 | 4.37 | 3.96 | 8.76 | 19.53 | 8.41 | 8.19 | 12.04 |
| Ours | **11.76** | **4.26** | 4.07 | **6.70** | **12.13** | **4.93** | 4.61 | **7.22** |

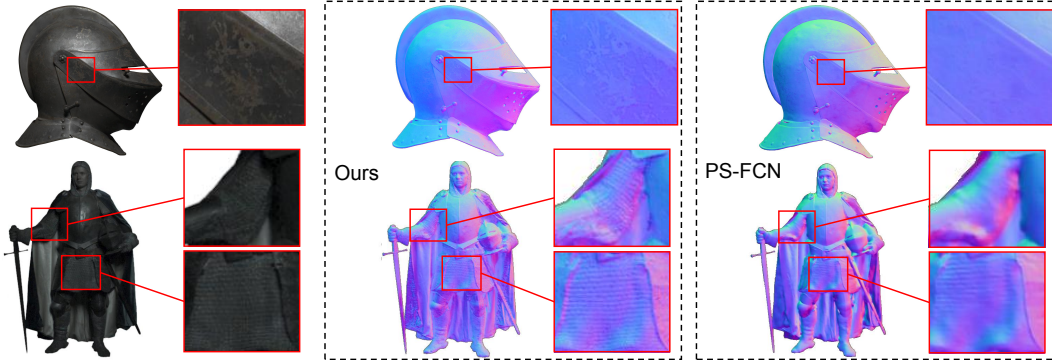

Figure 2: Qualitative results on the Light Stage Data Gallery dataset.

**Settings and implementation details.** Our framework is implemented in TensorFlow. We train our model using a batch size of 32 for 30 epochs, which takes about 10 hours using a single GeForce GTX 1080 Ti GPU. Images in the training dataset are randomly cropped and scaled to $32\times32$ to increase the training speed, and the testing is performed at the original resolution of input images. The learning rate is initially set to 0.01 and halved every 3 epochs. Adam optimizer is used to optimize our network with default parameters ($\beta_1 = 0.9$ and $\beta_2 = 0.999$). The widely used mean angular error (MAE, the lower the better) in degree is adopted to measure the accuracy of the estimated normal map. In all the experiments, we perform 100 random trials and average them.

### 5.2 Quantitative Comparison on the DiLiGenT Dataset

We compare GPS-Net with six state-of-the-art methods, including linear least squares based method (LS [1]), three per-pixel learning-based methods (CNN-PS [4], LMPS [5] and SPLINE-Net [6]), and two all-pixel learning-based methods (NEURAL-PS [30] and PS-FCN [7]). We perform the testing on the DiLiGenT benchmark [2] with seven sets of different numbers of input images (lighting conditions) varying from sparse to dense. Experimental results are shown in Table 1. First, we compare GPS-Net with per-pixel methods [4]–[6].[3] As analyzed in Section 4.1, the fixed-size observation map makes CNN-PS [4] difficult to find a balance between sparse and dense lighting distributions. To achieve good results under dense conditions, CNN-PS sets $w = 32$, which causes a significant performance drop as the number of lights decreases ($\leq 32$). Although CNN-PS achieves the best performance under dense lighting distributions (64 and 96), GPS-Net achieves better performance under sparse and moderate numbers of lights ($\leq 32$), because UFE-Layer is able to efficiently explore the inter-image variation given an arbitrary number of inputs. Although LMPS [5] and SPLINE-Net [6][4] are proposed to improve the performance of sparse observation maps, they cannot fundamentally solve its contradiction between resolution and density. Hence, GPS-Net outperforms them under both sparse and dense lighting distributions. Then, we compare GPS-Net with all-pixel methods [7], [30]. Due to

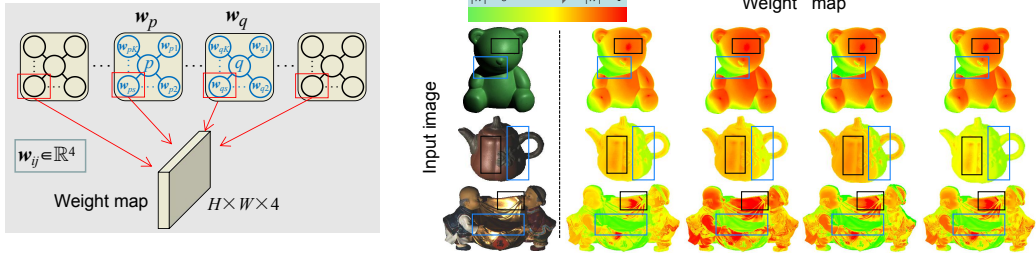

Figure 3: Visualization of the adaptive weight maps. The left shows how to visualize a weight map, and the right shows the four channels of the weight map. Black boxes mark specular highlights and blue boxes mark shadows.

Table 4: Effectiveness of Eq. (5).

|  | \multicolumn{7}{c}{Number of input images} | | | | | | | Avg. | Std. |
|  | 4 | 8 | 10 | 16 | 32 | 64 | 96 | Avg. | Std. |
|---|---|---|---|---|---|---|---|---|---|
| Max | 14.90 | 10.21 | 9.67 | 8.73 | 8.29 | 8.04 | 7.92 | 9.68 | 2.27 |
| Avg | 15.88 | 12.38 | 11.72 | 11.02 | 10.59 | 10.47 | 10.44 | 11.79 | **1.80** |
| Max+Avg | **13.46** | **10.07** | **9.43** | **8.71** | **8.05** | **7.84** | **7.81** | **9.34** | 1.86 |

the effectiveness of our multi-scale and multi-branch NR-Net and the ability to explore the additional inter-image lighting variation, GPS-Net achieves an all-round performance improvement compared to them. Overall, GPS-Net achieves the best average performance over existing methods, and more stable performance under varying lighting distributions than existing deep learning methods. Table 2 specifically shows the results on each object in the DiLiGenT dataset when the number of input images is 10 and 96. More quantitative results on each object with different numbers of inputs and visual comparisons of the predicted normal maps are shown in our supplementary material.

## 5.3 Results on the Synthetic Dataset

Table 3 shows the performance of GPS-Net, per-pixel method CNN-PS [4] and all-pixel method PS-FCN [7] on the synthetic dataset with 100 diverse BRDFs. In all the experiments, we perform 100 random trials and average them. It can be seen that GPS-Net achieves the best overall performance, which shows its robustness to different materials and varying input numbers.

## 5.4 Qualitative Results on Other Real-world Datasets

To further verify the capability of GPS-Net, we perform qualitative experiments on two challenging real datasets without ground truth, *i.e.*, the Light Stage Data Gallery dataset [36] and the Gourd&Apple dataset [22]. Fig. 2 shows the comparison results between GPS-Net and a typical all-pixel method PS-FCN [7] on two objects of the Light Stage Data Gallery dataset. More results on all the objects are shown in our supplementary material. As shown in Fig. 2, GPS-Net can recover pixel-wise surface normals in clearer details. This attributes to the fact that we adopt a multi-scale and multi-branch design in the NR-Net to avoid over-smoothing in the spatial domain, which is different from those all-pixel networks with all-$3\times3$ architectures [7], [8].

## 5.5 Network Analysis

**Learning proper weights for shadows and specular highlights.** As mentioned in Section 4.1, our SGC filters can learn to generate diverse weights for each pixel according to Eq. (7). This is different from traditional filters whose weights are the same for each pixel. We verify our adaptive weighting mechanism through visual experiments. As shown in Fig. 3, the generated weights that come from the same input image are arranged in pixel order, and then we can get a $4$-channel weight map corresponding to a certain input image. By visualizing it, we can observe what weights SGC filters will generate when there are outliers or useful clue observations in the input image. It is interesting to note that the learned weights share similar spirits with how non-learning approaches are

Table 5: Effectiveness of NR-Net architecture.

| Legend | MAE | | MAE |
|---|---|---|---|
| 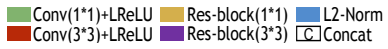 | | | |
|  | 9.39 [I] |  | 8.38 [II] |
|  | 7.90 [III] |  | **7.81** [IV] |

designed to improve their accuracies. The smaller weights usually appear in attached/cast shadow regions, indicating observations that should be discarded [14], [37]. The higher weights are given to pixels whose normals bisect the lighting and viewing directions, indicating "specular spikes" that easily tell the correct normals [23]. In those near-diffuse regions that produce near-Lambertian observations, SGC filters generate moderate weights. In this way, SGC filters appropriately balance the photometric stereo observations when extracting features from them.

**Effectiveness of Eq. (5).**    To verify the effectiveness of Eq. (5), we experimentally explore the effect of maximum and average operations in Eq. (5) in Table 4. Max-pooling achieves better performance than averaging (also demonstrated in Section 4.1 of PS-FCN [7]), while averaging is more robust to the varying input numbers. The combination of them achieves the best results.

**Effectiveness of NR-Net architecture.**    We verify the effectiveness of our multi-scale and multi-branch design in Table 5. We conduct the comparative experiments on the DiLiGenT benchmark [2] under all the 96 input images. The comparison between the results numbered by I, II, and III shows that, although $3 \times 3$ convolutional layers can explore spatial information to improve performance, too many of them will cause over-smoothing and performance degradation, and an appropriate number of them can make efficient use of the spatial information to achieve good performance. The result numbered by IV shows that our multi-scale and multi-branch design achieves the best performance.

# 6    Conclusion

We propose GPS-Net to combine the advantages of state-of-the-art learning-based frameworks by exploring both inter-image and intra-image variation of photometric stereo input. The UFE-Layer for per-pixel operation explores the inter-image intensity variation and appropriately balances it using SGC filters. The NR-Net for all-pixel operation makes efficient use of the intra-image spatial information to predict a high-resolution normal map. Experimental results indicate that GPS-Net achieves excellent performance under both sparse and dense lighting distributions. However, there are still some limitations in our method. We pay more attention on making our framework perform well with an arbitrary number of inputs that vary from sparse to dense. But we do not achieve the best accuracy under dense lighting distributions (*e.g.*, 96 input images). So we will investigate how we can make more efficient use of the dense photometric stereo observations in our future work.

# Broader Impact

The proposed framework will promote the development of photometric stereo technology, which can be useful for applications requiring 3D models with fine details (such as movie, game and other entertainment industries or industrial inspection). With the advancement of this technology, people may conveniently recover high-resolution and high-accuracy 3D information from images through specialist devices and efficient algorithms.

However, the efficient acquisition of high-quality 3D models (such as faces) may cause severer privacy violation problems than 2D images. We suggest that policymakers should establish an efficient monitoring platform to regulate the illegal spread of 3D models with private information that may cause ethical problems.

## Acknowledgments and Disclosure of Funding

This work was supported in part by Tianjin Research Program of Application Foundation and Advanced Technology (18JCYBJC19200), National Natural Science Foundation of China (61872012 and 61672096), National Key R&D Program of China (2019YFF0302902), and Beijing Academy of Artificial Intelligence (BAAI).

## Footnotes

[1]Throughout this paper, we assume the camera is radiometrically calibrated and the images are linearized, so we use "intensity" to refer to image irradiance for simplicity.

[2]In practice, the observed intensity $I$ is normalized by dividing the light source intensity in a pre-processing stage.

[3] The training datasets used by per-pixel methods [4]–[6] differ greatly from those of our GPS-Net and all-pixel methods [7], [8] in terms of rendering approach and data accuracy. So their performance drops severely when we retrain them using our training dataset.

[4] SPLINE-Net [6] only publishes its testing code and pre-trained model with an input number of 10 or less. For numbers of inputs larger than 10, *e.g.*, 16, we select 10 of 16 lights that have the least correlation in direction as the input and test it on the pre-trained model.

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
