[Supplementary Material]

# Supplementary Material for GPS-Net: Graph-based Photometric Stereo Network

**Zhuokun Yao**[1], **Kun Li**[1*], **Ying Fu**[2], **Haofeng Hu**[3], **Boxin Shi**[4,5*]
[1]College of Intelligence and Computing, Tianjin University, Tianjin, China
[2]School of Computer Science and Technology,
Beijing Institute of Technology, Beijing, China
[3]School of Precision Instrument and Opto-Electronics Engineering,
Tianjin University, Tianjin, China
[4]Department of Computer Science and Technology, Peking University, Beijing, China
[5]Institute for Artificial Intelligence, Peking University, Beijing, China
{yaozk,lik,haofeng_hu}@tju.edu.cn, fuying@bit.edu.cn, shiboxin@pku.edu.cn

In this supplementary material, we provide additional experimental results that are omitted in the paper due to page limitation. Specifically, we provide the following additional results.

1. Visual weight maps.

2. Qualitative results on the Light Stage Data Gallery dataset [1] and the Gourd&Apple dataset [2].

3. Quantitative results on each BRDF in the synthetic dataset.

4. Quantitative results and visual comparisons on the DiLiGenT benchmark [3].

## 1 Additional visual weight maps

In Fig. 1, we show more weight maps for additional scenes from the DiLiGenT dataset [3]. It can be seen from the figure that the SGC filters can generate larger weights for specular highlights that are important clues for normal estimation, and generate near-zero weights to suppress outliers in attached shadows and cast shadows.

## 2 Additional qualitative results

In Fig. 2, we show additional qualitative comparisons between our method and PS-FCN [4] on two challenging real-world datasets without ground truth, *i.e.*, the Light Stage Data Gallery dataset [1] and the Gourd&Apple dataset [2]. It can be seen from the figure that our method recovers pixel-wise surface normals in clearer details. This is because we adopt a multi-scale and multi-branch design in the NR-Net to avoid over-smoothing in the spatial domain, which is different from those all-pixel methods with all-$3\times3$ architectures [4], [5].

## 3 Additional results on each BRDF in the synthetic dataset

In Fig. 3, we show the comparisons between GPS-Net, CNN-PS [6] and PS-FCN [4] on each BRDF in the synthetic dataset. It can be seen from the figure that our method achieves excellent performance on diverse materials.

---

# 4    Additional experimental results on the DiLiGenT benchmark

We perform quantitative experiments and visual comparisons [3] between our method and five state-of-the-art methods including LS [7], CNN-PS [6], SPLINE-Net [8], NEURAL-PS [9] and PS-FCN [4] on the DiLiGenT benchmark.[2] First, We show their performance on each object in the DiLiGenT dataset under diverse lighting distributions in Fig. 4 and Fig. 5. Then in Fig. 6-Fig. 40, we show the visual results of their predicted normal maps and error maps. The values in these figures represent mean angular errors (MAEs, the lower the better). In all the experiments, we perform 100 random trials and average the results. It can be seen from these experimental results that our method achieves high-accuracy normal prediction under both sparse and dense lighting distributions.

## Footnotes

[2]The code for LMPS [10] has not been published online. We contacted the authors to provide us the values on benchmark data.

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

Figure 1: Additional visual weight maps. All the four channels of each weight map are shown. Black boxes mark specular highlights and blue boxes mark shadows.

Figure 2: Additional Qualitative Results on the Light Stage Data Gallery dataset [1] and the Gourd&Apple dataset [2].

Figure 3: Results on the synthetic dataset with different BRDFs.

Figure 4: Quantitative comparisons on each object in the DiLiGenT dataset when the number of inputs is 4, 8, 10 and 16.

Figure 5: Quantitative comparisons on each object in the DiLiGenT dataset when the number of inputs is 32, 64 and 96.

Figure 6: Visual comparisons of "ball" and "cow" under 4 lightings.

Figure 7: Visual comparisons of "bear" and "cat" under 4 lightings.

Figure 8: Visual comparisons of "pot1" and "pot2" under 4 lightings.

Figure 9: Visual comparisons of "buddha" and "goblet" under 4 lightings.

Figure 10: Visual comparisons of "reading" and "harvest" under 4 lightings.

Figure 11: Visual comparisons of "ball" and "cow" under 8 lightings.

Figure 12: Visual comparisons of "bear" and "cat" under 8 lightings.

Figure 13: Visual comparisons of "pot1" and "pot2" under 8 lightings.

Figure 14: Visual comparisons of "buddha" and "goblet" under 8 lightings.

Figure 15: Visual comparisons of "reading" and "harvest" under 8 lightings.

Figure 16: Visual comparisons of "ball" and "cow" under 10 lightings.

Figure 17: Visual comparisons of "bear" and "cat" under 10 lightings.

Figure 18: Visual comparisons of "pot1" and "pot2" under 10 lightings.

Figure 19: Visual comparisons of "buddha" and "goblet" under 10 lightings.

Figure 20: Visual comparisons of "reading" and "harvest" under 10 lightings.

Figure 21: Visual comparisons of "ball" and "cow" under 16 lightings.

Figure 22: Visual comparisons of "bear" and "cat" under 16 lightings.

Figure 23: Visual comparisons of "pot1" and "pot2" under 16 lightings.

Figure 24: Visual comparisons of "buddha" and "goblet" under 16 lightings.

**16 inputs**

Ground Truth

Ground Truth

| | | | | |
|---|---|---|---|---|
| **Ours** | | 13.30° | | 16.14° |
| **LS** | | 19.49° | | 31.15° |
| **CNN-PS** | | 12.17° | | 17.04° |
| **SPLINE-Net** | | 16.33° | | 18.37° |
| **NEURAL-PS** | | 12.63° | | 25.42° |
| **PS-FCN** | | 13.93° | | 17.16° |

0°                                                                          90°

Figure 25: Visual comparisons of "reading" and "harvest" under 16 lightings.

Figure 26: Visual comparisons of "ball" and "cow" under 32 lightings.

Figure 27: Visual comparisons of "bear" and "cat" under 32 lightings.

Figure 28: Visual comparisons of "pot1" and "pot2" under 32 lightings.

Figure 29: Visual comparisons of "buddha" and "goblet" under 32 lightings.

Figure 30: Visual comparisons of "reading" and "harvest" under 32 lightings.

Figure 31: Visual comparisons of "ball" and "cow" under 64 lightings.

Figure 32: Visual comparisons of "bear" and "cat" under 64 lightings.

Figure 33: Visual comparisons of "pot1" and "pot2" under 64 lightings.

Figure 34: Visual comparisons of "buddha" and "goblet" under 64 lightings.

Figure 35: Visual comparisons of "reading" and "harvest" under 64 lightings.

Figure 36: Visual comparisons of "ball" and "cow" under 96 lightings.

Figure 37: Visual comparisons of "bear" and "cat" under 96 lightings.

Figure 38: Visual comparisons of "pot1" and "pot2" under 96 lightings.

Figure 39: Visual comparisons of "buddha" and "goblet" under 96 lightings.

Figure 40: Visual comparisons of "reading" and "harvest" under 96 lightings.