[Reviews · NeurIPS 2020]

Review 1

Summary and Contributions: The paper proposes a new deep learning-based photometric stereo method, that uses a graph-based approach to aggregate information from multiple observations at each pixel to generate a feature map, which is then used to incorporate spatial information and regress to the final output. --Post-rebuttal I've read the rebuttal, but unfortunately, it doesn't adequately address my concerns: 1. The new table doesn't really answer that both R2 and I brought up. In some sense, there's no real apples-to-apples comparison: - It shows its own ablation of SM+3x3 CNN does about 0.4 degrees worse than UFE+3x3 CNN. But the 3x3 CNN architectures used in these two seem to be different. - Moreover, PS-FCN [7]'s SM+3x3 FCN (which actually features a different 'SM' architecture) does equivalently to UFE+3x3 CNN. - They also present new results for UFE+3x3 FCN, which does do better than PS-FCN[7] (but by a smaller amount of 0.2 degrees), but the FCN architecture used here is not the same as the FCN architecture used in PS-FCN! A far more logical experiment would have been to compare to SM+NR-net (using the PS-FCN SM architecture if possible). This would conclusively show the separate contributions of the UFE layer and NR-net to the model's performance over previous all-pixel methods (specifically, PS-FCN). 2. The authors explain that the observation-map approach to single pixel processing performs worse with sparse inputs. I believe that, but the question is whether it does better than the graph-based UFE layer and helps in avoiding a 'resolution trade-off'. For evidence towards this, the rebuttal points to the fact that CNN-PS does worse than the proposed method for sparse inputs (but note, it actually does *better* for denser inputs with >= 64 images). But the problem is that the CNN-PS does no spatial processing at all! So it isn't clear to me whether the proposed method outperforms CNN-PS because the UFE layer is better than observation maps, or because of the NR-net which is using spatial processing and likely reasoning with smoothness across pixels. 3. Regd. the rebuttal response for "R1: The suggested ablation study (Observation maps + Pooling + NR-Net)", I made a mistake in including pooling in that suggestion in the original review. What I meant to say was to use an observation-map based approach to generate a feature vector for each pixel, and then feed that feature map into NR-net (so no pooling's required as the rebuttal points out). The point is to swap out the UFE layer with an observation map-based layer in the proposed method. 4. The response to the ablation study on equation (5) actually brings up new questions. The rebuttal says that doing only max- or avg- pooling in (5) is worse than doing both (because each has its own benefits). Seeing the quantitative results of that evaluation are important---to see how the proposed layer with only max-pooling in equation (5) compares to PS-FCN which also does only max-pooling (as do all the "SM" blocks in the new results table in the rebuttal). If they're equivalent, this would bring up the question whether the benefit of UFE over these past methods is simply because of better (i.e., both max + avg) pooling. I think it may indeed be true that the UFE layer is a useful and better way of reasoning with photometric inputs---which would be an interesting and novel contribution. But this needs to be demonstrated with the right ablation studies (many of which understandably could not be run within the short time-frame of the rebuttal period). As things stand, I'd prefer that the paper be re-reviewed with these studies before being accepted.

Strengths: - The idea of treating multiple observations at each pixel as nodes in a graph is interesting.

Weaknesses: Broadly, the paper makes a number of claims while motivating the work that do not seem to be backed up. At the same time, its experimental improvement is relatively minor, and it is not clear whether even these improvements come from its graph formulation. - While the main contribution of the paper is arguably in its graph-based "UFE-layer", it appears from the ablation study in Table 3 that most of its benefits over previous 'all pixel' method [7] comes from its NR-net architecture. This largely undercuts not only the specific graph-based design of the UFE-layer, but also the more general claim that 'all-pixel' methods do not make full use of 'inter-image' information. - In defining the UFE-layer as graph-based aggregation, the paper presents the motivation that an 'observation map' type approach as used in older single-pixel methods (like [4]) is sub-optimal, because it requires a resolution trade-off. This is not experimentally demonstrated—and should be, because it is central to the whole 'graph-based' premise. An important ablation would be to use an observation map-based layers (like [4]) as an alternative to the UFE-layers, which generates feature maps that are then pooled and fed to NR-net. If such an approach works just as well or better, it would significantly diminish the contribution and narrative of this paper. - It is worth noting that the single pixel method [4] actually does better than the proposed method with larger numbers of inputs (Table 1: for 64, 96). This would suggest that with more inputs, when the 'intra-image' information is stronger, the proposed method isn't able to exploit it as well as [4]. - Beyond the way that the UFE-layer generates its per-image feature vectors (by considering all other images), another difference to standard all-pixel methods is the nature of pooling. Other all-pixel methods just use max-pooling, while the proposed method seems to use both average and max-pooling (equation 5). It is not clear to what extent this contributes to the advantage of the proposed method over previous all-pixel approaches, and should be verified by running an ablation with only max-pooling in equation 5.

Correctness: See above.

Clarity: Yes, although I recommend a larger / separate figure to explain the UFE layer.

Relation to Prior Work: Mostly, but more ablations/comparisons are needed (see above).

Reproducibility: Yes

Additional Feedback:


Review 2

Summary and Contributions: This paper introduces a graph-based network called GPS-Net for the problem of calibrated photometric stereo. An unstructured feature extraction layer is proposed to exploit inter-image information by arranging multiple observations of a pixel into a graph structure and applying structure-aware graph convolution to extract a fixed-size feature. A normal regression network is proposed to make efficient use of intra-image spatial information by arranging the per-pixel features into a structured feature map and regressing a high-resolution and high-accuracy normal map. The proposed method takes advantages of both per-pixel approach and all-pixel approach, and shows promising results on three benchmarks.

Strengths: + The idea of using structure-aware graph convolution to handle an arbitrary number of unorder images sounds novel. It helps to avoid the trade off between resolution and density in using observation maps. + The proposed normal regression network has a multi-branch design with a 1x1 conv branch and a 3x3 conv branch. This helps to produce shaper normal map with finer details than previous method PS-FCN using only 3x3 conv layers. + The adaptive weight maps visualized in Figure 3 help to understand the behavior of the proposed unstructured feature extraction layer. + The proposed method achieves robust results on the DiLiGenT benchmark under both sparse and dense lighting distribution. Qualitative results on Light Stage Data Gallery and Gourd and Apple dataset are promising.

Weaknesses: - The authors claim that the proposed structure-aware graph convolution filter is helpful. However, no ablation study is done to verify the design. What is the result if we just use a simple graph convolution layer? - The effect of including the “maximum convolution result” in Equation 5 is unclear. Ablation study is needed to support such a design. - The major evaluation is done on 10 objects illuminated under a fixed lighting distribution from the DiLiGenT benchmark, which is rather limited. The authors should evaluate their method on a wide diversity of synthetic data (e.g., different shapes rendered using different BRDFs under different (biased) lighting distributions) to justify the proposed method. - It is unclear what is the input image number during training of the proposed method. In Table 1, does the training number equals to the testing number? It would be helpful to analyze the effect of training image number as well. - It would be desirable to include another baseline network using only 1x1 convolutions in Table 3 for comparison. To make a fair comparison, different networks with a similar number of parameters should be compared. - It would be desirable to include runtime comparison of different deep learning based methods.

Correctness: The proposed method sounds correct. The ablation study, however, is not very comprehensive.

Clarity: Overall the paper is well written. Further elaboration on how multi-scale implementation is achieved for NR-Net is needed though.

Relation to Prior Work: References to prior work are sufficient and the contributions made in this paper are clearly described.

Reproducibility: Yes

Additional Feedback: Figure 1 seems a bit messy, especially the arrows originating from the middle part.


Review 3

Summary and Contributions: This paper presents a novel network design for photometric stereo that explores both inter-image and intra-image information for normal estimation. The inter-image part is based on a graph network that can deal with an arbitrary number of input images. This part is inspired by the work in [31] and novel. Later, a feature vector is produced at each pixel. These features at different pixels are arranged to according to the pixels' image coordinate to produce a tensor. This tensor is further sent to a CNN to exploit intra-image information. The main contribution of the paper is that it is the first time to explicitly extract intra-image and inter-image information and fuse them for photometric stereo. I regard this as the biggest contribution which makes this paper stands out from the others.

Strengths: The paper is very well written and easy to follow. The network structure is quite novel and makes sense. The subnetworks for iter-image and intra-image information extraction complement each other. It is the first time that both inter-image and intra-image information are fused to solve photometric stereo. Experiment results are reasonable on the DiLiGent dataset, though not significantly better than previous methods. Overall, I still like this paper for its novelty in network architecture (please see my earlier comments in the summary). It is not perfect yet, as the weakness I mentioned below. But it is a clear acceptance to me.

Weaknesses: While the network is quite novel, the experiment results are not significantly better than previous methods. According to Table 2, when there are 10 input images, CNN-PS[4] performs best at most of the time. When there are 96 input images, SPLINE-Net[6] and NEURAL-PS[29] are strong competitors. But this method has the best overall performance under different numbers of input images, it is also an important advantage.

Correctness: It looks correct.

Clarity: Yes, it is clearly written and nicely motivated.

Relation to Prior Work: Yes.

Reproducibility: Yes

Additional Feedback: Update After Rebuttal The experiments are somewhat weak that the improvement over previous methods is not too significant. This suggests the UFE-layer is perhaps not that effective. But I still like the idea of combing inter-image and intra-image information and believe it could inspire following up works on this direction. I still recommend to accept it.


Review 4

Summary and Contributions: This paper proposed a graph-based photometric stereo network which uses graph structure to model the variable number of inputs in photometric stereo. The proposed network consists of two sub-network: GPS-network to estimate the weights parameter for feature aggregation, and NR-Net to estimate the final results with spatial relationship consideration through CNN architecture. The proposed method is compared with several state-of-the-art methods and demonstrated better performance.

Strengths: The proposed GPS-Net is effective in aggregating observations from variable number of inputs. The performance is state-of-the-art.

Weaknesses: The contribution of NR-Net is weak as it is only a standard CNN network for "post-processing" after the GPS-Net. The Qualitative comparisons in Fig. 2 is biased as it only compare with PS-FCN which is not the most state-of-the-art methods, instead it should compare with LMPS and Neural-PS which are more recent and representative. The quantitative comparison actually reveal that the proposed method is not as good as CNN-PS if there is large number of inputs and it is also not as good as Neural-PS if the number of inputs is small. The proposed network is only trained on synthetic dataset, and the number of examples are limited in the synthetic dataset. Although the paper has demonstrated some real-world examples, there are only 10 real-world examples which is quite limited.

Correctness: They looks ok to me.

Clarity: Yes

Relation to Prior Work: Yes

Reproducibility: Yes

Additional Feedback: I think the contribution of this paper is marginal. I appreciate the proposed GPS-Net in handling variable number of inputs and agree that this method is better than the max pooling used in PS-FCN in aggregating features from multiple inputs. However, I also feel that the experiments are insufficient (please check my comments in paper weakness) especially for the qualitative comparisons. Since the number of training and testing data are small, it would be easy to overfit the data, and I feel this is the purpose of the NR-Net which is to rectify the errors from the GPS-Net.

[Author Response · NeurIPS 2020]

1 We sincerely thank all reviewers for their valuable suggestions. Below we respond to the comments and concerns.

2 **R1&R2: Effectiveness of UFE-layer.** We use the fixed network architecture to further demonstrate the effectiveness of UFE-Layer in the table below. We also add the $1\times1$ network suggested by R2 in the table. For convenience, we use "SM" to represent the "shared-weight feature extractor + max-pooling" strategy of PS-FCN [7]. The comparison between SM+3×3 CNN and UFE-Layer+3×3 CNN and the comparison between SM+3×3 FCN and UFE-Layer+3×3 FCN prove the effectiveness of UFE-Layer. Although the first row of Table 3 (UFE-Layer+3×3 CNN) achieves almost the same performance as PS-FCN (SM+3×3 FCN), it does not mean that UFE-Layer has no contribution compared to SM. PS-FCN uses a Fully Convolutional Network (3×3 FCN), and part of the 3×3 layers have strides as 2 to achieve down-sampling and up-sampling. We think that setting strides as 2 has a similar effect to the $1\times1$ layers in our NR-Net, *i.e.*, weakening the mutual influence among pixels and reducing over-smoothing in the spatial domain. As shown in the table below, our UFE-Layer+NR-Net achieves the best performance.

| Method | Diagram | Avg. | Method | Diagram | Avg. |
|---|---|---|---|---|---|
| | Conv(1*1,stride=1)+LReLU · Res-block(1*1) · L2-Norm · Conv(3*3,stride=2)+LReLU · Conv(3*3,stride=1)+LReLU · Res-block(3*3) · Concat · Deconv(3*3,stride=2)+LReLU | | | | |
| SM+3×3 CNN | Shared-weight feature extractor (Input 1 … Input K) → Max | 8.79 | SM+3×3 FCN (PS-FCN [7]) | Shared-weight feature extractor (Input 1 … Input K) → Max | 8.39 |
| UFE-Layer+3×3 CNN (The fist row in Table 3) | UFE-Layer (Input 1 … Input K) | 8.38 | UFE-Layer +3×3 FCN | UFE-Layer (Input 1 … Input K) | 8.20 |
| UFE-Layer+NR-Net (The second row in Table 3) | UFE-Layer (Input 1 … Input K) | 7.81 | UFE-Layer +1×1 CNN | UFE-Layer (Input 1 … Input K) | 9.39 |

12 **R1&R3&R4: The resolution trade-off in single-pixel methods.** CNN-PS [4] states:"The size of the observation map ($w$) should be chosen carefully. As $w$ increases, the observation map becomes sparser. On the other hand, the smaller observation map has less respresentability." The later methods based on observation maps [5, 6] also clearly indicate: When the input number decreases, it causes serious sparsity problem and performance degradation, which is an obvious manifestation of resolution trade-off. Therefore, CNN-PS only conducts experiments under dense inputs (96) by setting a large $w$ (32). However, its performance drops significantly under sparse inputs, as shown in Table 1. LMPS [5] sets $w$ as 14 to achieve better performance under sparse inputs, but its performance under dense inputs has declined. This clearly reflects that it is not easy to take the resolution trade-off. Although GPS-Net is not as good as CNN-PS under dense inputs, we achieve the best results when the input number is less than 64, and the best overall performance, as shown in Table 1.

22 **R1: The suggested ablation study (Observation maps + Pooling + NR-Net).** We respectfully point out that we should not fuse the feature maps generated by observation maps through any pooling operation (like PS-FCN [7]). Because the max-pooling [7] is performed for feature maps under different lightings, but the feature maps of observation maps are generated for each pixel, with varying lighting information encoded in the observation maps.

26 **R1&R2: The ablation study of Eq. (5).** Due to the limited rebuttal time, we only quickly test the effect of max-pooling and averaging using a smaller dataset. We find that max-pooling achieves better performance than averaging (also demonstrated in Section 4.1 of PS-FCN [7]), while averaging is more robust to the varying input numbers. The combination of them achieves the best results with robustness. We are conducting the complete test and will add the detailed ablation study in the final version.

31 **R2: The ablation study of SGC filters.** The general spectral graph convolution networks require the input graphs to have the same topologies, *i.e.*, the fixed input number during training and testing, which is similar to DPSN [3]. Like the state-of-the-art methods [4-7,29], we aim to handle an arbitrary number of inputs, and hence we use our SGC filters to handle graphs containing an arbitrary number of adjacent nodes (graphs with inconsistent topologies).

35 **R2: The training number.** To show the flexibility of our model, we did not use the same training number as the testing number. We have trained three models under 4, 8 and 16 inputs to test under {4}, {8,10} and {16,32,64,96} inputs, respectively.

38 **R4: The contribution of NR-Net.** We respectfully point out that GPS-Net is our entire network including UFE-Layer and NR-Net. The contribution of NR-Net is demonstrated by the comparison with "all-pixel" methods [7,29]. Qualitative results in Figure 2 (paper) and Figure 2 (supplement) illustrate that NR-Net can predict normal maps with richer details. It is quite important in photometric stereo that aims at acquiring high-resolution 3D information [1].

42 **R4: The qualitative comparisons in Figure 2.** The qualitative comparisons with state-of-the-art methods including NEURAL-PS [29] were shown in Figures 5-39 in the supplementary material. The code for LMPS [5] has not been published online. Hence, we contacted the authors, and they were only able to provide the numbers on benchmark data. Figure 2 is just an example to show the contribution of NR-Net in preserving high-resolution details compared with "all-pixel" methods [7,29]. Hence, we chose the best-performing all-pixel method PS-FCN [7] in Table 1 as a representative for comparison. Figure 2 in the supplementary material gives more results.

48 **R1&R2&R4: Other suggestions.** We will make the following changes as suggested in the final version. 1) We will separate Figure 1 and show it in a clear way (R1, R2). 2) Since DiLiGenT [2] is currently the only real-world photometric stereo benchmark, we will try our best to test GPS-Net on more synthetic data with diverse BRDFs and shapes (R2, R4). 3) We will make a runtime comparison and show how multi-scale implementation is achieved for NR-Net (R2).

[Meta-Review · NeurIPS 2020]

This paper was reviewed by four reviewers knowledgeable in photometric stereo. It received scores of 4,7,8, and 6. After reading the rebuttals and each other's reviews, none of the reviewers changed their opinion. There was a lot of discussion about this paper. R1 has some valid concerns about the ablation experiments. This has been partially resolved thanks to the additional experiments in the rebuttal, but R1 still had some remaining doubts. R2, R2, and R4 were more positive and felt that this work might open the door to further exploration of hybrid photometric approaches that combine per-pixel and all-pixel reasoning. They also commented on the qualitative and quantitative performance. Despite R1's reservations, this AC sides with the other reviews and thinks there is enough merit in accepting the paper. For the final paper, the authors are strongly encouraged to (i) add the ablation experiments from the rebuttal, (ii) improve the presentation of Fig 1 (see R1 and R2's comments), and (iii) add the ablation of Eq 5 promised to R1 and R2. Furthermore they should take the detailed comments of R1 into account, as they will further strengthen the paper.